# Polyphenols-Rich Fraction from *Annona muricata* Linn. Leaves Attenuates Oxidative and Inflammatory Responses in Neutrophils, Macrophages, and Experimental Lung Injury

**DOI:** 10.3390/pharmaceutics14061182

**Published:** 2022-05-31

**Authors:** André Lopes Saraiva, Allisson Benatti Justino, Rodrigo Rodrigues Franco, Heitor Cappato Guerra Silva, Felipe dos Santos Arruda, Sandra Gabriela Klein, Mara Rúbia Nunes Celes, Luiz Ricardo Goulart, Foued Salmen Espindola

**Affiliations:** 1Institute of Biotechnology, Federal University of Uberlândia, Rua Acre s/n, Bloco 2E, Uberlândia 38400-902, MG, Brazil; allssaraiva@gmail.com (A.L.S.); allissonbjustino@hotmail.com (A.B.J.); rodrigorfr@yahoo.com.br (R.R.F.); heitorcappato@gmail.com (H.C.G.S.); 2Department of Bioscience and Technology, Institute of Tropical Pathology and Public Health, Federal University of Goiás, Rua 235, Setor Leste Universitário, Goiânia 74605-050, GO, Brazil; felipefagote.arruda@gmail.com (F.d.S.A.); rubia.celes@gmail.com (M.R.N.C.); 3Rodent Vivarium Network (REBIR), Dean of Research and Graduate Studies, Federal University of Uberlândia, Rua Ceará s/n, Bloco 4U, Uberlândia 38405-315, MG, Brazil; sandragabrielaklein@gmail.com

**Keywords:** Annonaceae, natural products, neutrophils, macrophages, reactive oxygen species, acute lung injury

## Abstract

*Annona muricata* Linn. is a common plant found in the warmest regions of South and Central America and its use in traditional medicine has been reported for the treatment of various illnesses. In the current study, we investigate the antioxidant and anti-inflammatory activities of crude extract and fractions from *A. muricata* L. leaves in isolated murine phagocytic immune cells as well as experimental LPS-induced acute lung injury (ALI). In a luminol-dependent chemiluminescence assay, we showed that ethyl acetate (EtOAc.f) and n-butanol (BuOH.f) fractions—both rich in polyphenols—reduced the generation of reactive oxygen species (ROS) by neutrophils stimulated with opsonized zymosan; similar results were found in culture of bone marrow-derived macrophages (BMDMs). By evaluating anti-inflammatory activity in BMDMs, EtOAc.f and BuOH.f reduced secretion of IL-6 and expression of the co-stimulatory molecule CD40. Furthermore, in LPS-induced ALI, oral administration of EtOAc.f reduced myeloperoxidase (MPO) activity in lung tissue. In addition, on a mechanism dependent on glutathione levels, the oxidative damage was also attenuated. These findings revealed direct antioxidant and anti-inflammatory activities of polyphenols-rich fractions of *A. muricata* L. leaves on neutrophils and macrophages. Moreover, the reduced oxidative damage and levels of inflammatory markers in experimental ALI suggest that these fractions might be explored for the development of new therapies for inflammatory conditions.

## 1. Introduction

*Annona muricata* Linn., also known as soursop (English), graviola (Portuguese), and guanabana (Latin American Spanish), is a plant belonging to the Annonaceae family and its distribution covers Western Africa, Southeast Asia, and tropical regions of Central and South America [1]. For the treatment of a variety of illnesses, pharmacological activities and traditional medicinal uses of *A. muricata* have been reported, and include hypertension as a complication of diabetes [2], bacterial infections [3], and respiratory conditions [4]. In Brazil, the plant is used for weight loss [5] and in ethnoveterinary medicine as a treatment for snakebite [6].

Although all parts of the *A. muricata* are used, the leaves have received particular attention, and phytochemical investigations of leaf extracts have identified numerous bioactive compounds. Indeed, antioxidant activities have been associated with phytochemicals present in the essential oil of the leaves [7] and with phenolic compounds such as rutin, naringenin, and vanillin, which are the major constituents of the ethanolic extract [8]. In addition to antioxidant properties, effects on reducing inflammatory mediators such as cytokines and nitric oxide [9,10] have also evidenced anti-inflammatory actions of extracts of *A. muricata* leaves. However, studies addressing their antioxidant and anti-inflammatory effects directly on primary cultures of phagocytic immune cells such as neutrophils and macrophages are nonexistent. This opens up new possibilities for studying the effects of *A. muricata* leaf extracts since oxidative and inflammatory activities of neutrophils and macrophages have been implicated in a variety of inflammatory conditions, with a pivotal role in lung inflammation [11].

The lungs are in constant contact with the external environment, and in homeostasis, lungs must maintain a healthy state and avoid inflammatory stimulus such as allergens and pathogens. However, a wide range of direct and indirect insults can induce acute lung injury (ALI), and its most severe form, acute respiratory distress syndrome (ARDS). ALI/ARDS is characterized by intense inflammatory response leading to disseminated injury to the alveolar-capillary barrier, and accumulation of protein-rich edema fluid in airspaces, all contributing to severe gas exchange abnormalities [12]. In the management of the disease, respiratory support is the main intervention, and to date, no pharmacological treatment has been effective in reducing the mortality rate [13]. In this context, natural products are a valuable opportunity to identify bioactive compounds which could be used in the management of inflammation during ALI/ARDS [14]. Furthermore, while the traditional use of *A. muricata* in the treatment of respiratory conditions has been described [4], scientific data characterizing antioxidant and anti-inflammatory effects of this plant in experimental lung inflammation are still poorly defined

Thus, in the present study, we characterize the antioxidant and anti-inflammatory effects of polyphenols-rich ethyl-acetate and n-butanol fractions from *A. muricata* leaves on neutrophils and macrophages. Then, based on recent reports that reinforce that regulating the function of said cells may be a promising strategy against ALI/ARDS, we demonstrate the effects of fractions in a murine model of LPS-induced acute lung injury (ALI) that also involves both oxidative and inflammatory components.

## 2. Material and Methods

### 2.1. Reagents

Dried leaves of *A. muricata* were purchased (batch number 058671) from the pharmaceutical supply distributor Florien^®^ (São Paulo, Brazil). Fetal bovine serum (FBS), penicillin, streptomycin, and L-glutamine were acquired from Thermo Fisher Scientific (Gibco™, Waltham, MA, USA). Flow cytometry antibodies were acquired from BD Bioscience (Franklin Lakes, NJ, USA). When not mentioned, all other reagents were purchased from Sigma-Aldrich (St. Louis, MO, USA).

### 2.2. Animals

Male C57BL/6 mice, 6–8 weeks-old and specific pathogen free (SPF), were used. The animals were housed in a temperature-controlled room (23–25 °C) under a clear-dark cycle of 12 h and fed ad libitum. All the research conducted on the animals were performed under ARRIVE guidelines and previously approved by the Animal Ethics Committee of the Federal University of Uberlandia (protocol 079/18).

### 2.3. Plant Extracts and Fractions

We used the extracts from *A. muricata* L. leaves employed in previously published work from our group [15]. Briefly, phytoconstituents present in dried leaves of *A. muricata* were extracted through liquid-liquid partitioning. Dried leaves (1.0 Kg) were mechanically macerated in ethanol 98% (5.0 L) and the solution was kept for 3 days. Then, the solution was filtered and the ethanol was removed in a rotary evaporator under reduced pressure (40 °C). All water content was removed through lyophilization given rise crude extract (EtOH.cr; extract yield of 12.5% of the initial weight of dried leaves). Next, EtOH.cr was diluted in methanol/water solution (9:1 *v*/*v*, 200 mL), and in ascending order of polarity, solvent-solvent partition was carried out resulting in the following fractions (4 × 200 mL for each solvent): hexane fraction, dichloromethane fraction (DCM.f), ethyl acetate fraction (EtOAc.f), n-butanol fraction (BuOH.f), and water/aqueous fraction (Wa.f). To completely remove the volatile solvent and water from non-volatile compounds, rotary evaporation and lyophilization were also performed.

### 2.4. HPLC-ESI/MS^n^ Characterization of Phytoconstituents in EtOAc.f

The chemical compounds present in EtOAc.f and BuOH.f from *A.muricata* leaves were identified using High-Performance Liquid Chromatography-Electrospray Ionization-Tandem Mass Spectrometry (HPLC-ESI/MS^n^) as described and published by us [15]. An improved description of the method, a representation of the chromatogram, and the list of identified compounds are available in a referenced table in the Appendix A.

### 2.5. Primary Cell Culture

Both neutrophils and macrophages were obtained from bone marrow cells. For this, the mice were euthanized by cervical dislocation under anesthesia, and femur and tibia bones were dissected. For neutrophil isolation, bone marrow cells were aseptically harvested by flushing the medullary cavity using a needle and syringe filled with Hank’s Balanced Salt Solution (HBSS) without calcium; HBSS was replaced by RPMI-1640 for generation of bone marrow-derived macrophages (BMDMs).

Neutrophils were isolated by density gradient centrifugation. In brief, bone marrow cells suspended in 2 mL of HBSS were carefully layered upon Percoll gradient (3 mL of Percoll 65% and 3 mL of Percoll 72%) and subsequently centrifuged at 1.200× *g* for 30 min at 25 °C; neutrophil layer was aspirated from the interface between Percoll 65% and 72%.

BMDMs were generated by incubating bone marrow cells with RPMI-1640 supplemented with FBS (20%), penicillin (100 U/mL), streptomycin (100 µg/mL), L-glutamine (2 mM), and 30% of L929 cell-conditioned medium (LCCM), which was a source of macrophage colony-stimulating factor (m-CSF). BMDMs were isolated by removing supernatant of cell culture and adherent cells were detached with sterile cold PBS. LCCM was used only to obtain differentiated macrophages. In all other experiments, macrophages were incubated in LCCM-free medium.

### 2.6. ROS Production by Neutrophils and Macrophages

ROS production by neutrophils and BMDMs were detected by luminol-dependent chemiluminescence assay as previously described with few modifications [16]. The cells were suspended in HBSS without phenol red and seeded (2 × 10^5^ cells/well) in a 96-well white plate (OptiPlate^TM^, Perkin Elmer^®^, Waltham, MA, USA). The cells were pretreated with different fractions from *A.muricata* leaves for 30 min at 37 °C and 5% of CO_2_ atmosphere. For the concentration-response curve, we used 1, 3, and 10 µg/mL of crude extract and each fraction; subsequent experiments were performed with 3 µg/mL. Next, the cells were incubated with luminol (500 µM) for 10 min and then stimulated with zymosan (100 µg/mL) previously opsonized as described [17]. Specifically, in macrophages ROS production was also monitored under phorbol 12-myristate 13-acetate (PMA 100 ng/mL) stimulation. Chemiluminescence emission, resulting from the reaction between ROS and luminol, was monitored in a microplate reader (EnSpire Multimode Plate Reader™, Perkin Elmer^®^, Waltham, MA, USA) for an additional 60 min. The results are showed as Relative Luminescence Units (RLU).

### 2.7. Cell Viability of Neutrophils

Neutrophil viability under treatment with fractions from *A. muricata* was assessed using Fixable Viability Dye eFluor^®^ 780 (Invitrogen™, Waltham, MA, USA). The neutrophils were incubated with EtOAc.f and BuOH.f for 30 min and stained with anti-mouse Ly6G and dye viability following the manufacturer’s instructions. Through flow cytometry, the percentage of living neutrophils (Ly6G^+^/viability dye^−^) was determined. Untreated cells were considered 100% of viability.

### 2.8. BMDMs Phagocytosis Analysis

The phagocytic activity of BMDMs was determined by evaluating the ingestion of FITC (fluorescein isothiocyanate) pre-labeled zymosan (Zy-FITC). Preparation of Zy-FITC, as well as the phagocytosis assay, was carried out according to a previous report [17]. For the phagocytosis assay, 5 × 10^5^ BMDMs were pretreated with EtOAc.f and BuOH.f and then incubated for 30 min, at 37 °C, with Zy-FITC (100 µg/mL). Next, the cells were quickly placed on ice to stop the phagocytosis process. Cells were stained with anti-mouse CD11b antibody conjugated to a fluorochrome different from FITC. Frequencies of CD11b^+^/Zy-FITC^+^ cells were determined through flow cytometry (FACS Canto™, BD Biosciences, Franklin Lakes, NJ, USA).

### 2.9. Evaluation of Cytokine Secretion, Co-Stimulatory Molecules Expression, and Cell Viability Assay in BMDMs

BMDMs were seeded at a density of 2 × 10^5^ cells/well in a flat-bottom 96-well microplate (Corning^®^, New York, NY, USA). The cells were pretreated for 30 min with EtOAc.f and BuOH.f and stimulated with lipopolysaccharide (LPS; 1 µg/mL) from Gram-negative bacteria. After 24 h, the supernatant was harvested, and levels of IL-6 and TNF-α were determined by ELISA assay kit following the manufacturer’s instructions (DuoSet ELISA, R&D Systems). For analysis of expression of co-stimulatory molecules, 5 × 10^5^ cells were submitted to the same protocol for stimulation and then stained with anti-mouse F4/80, CD40, and CD80 for flow cytometry analyses. Expression of CD40 and CD80 was examined in F4/80 positive cells.

Cytotoxic effects of fractions were monitored in MTT cell viability assay. For this, after pretreatment with EtOAc.f and BuOH.f, 100 µL of MTT solution (3-(4,5-dimethylthiazolyl-2)-2,5-diphenyltetrazolium bromide) at 5 mg/mL was added to each well containing 2 × 10^5^ cells/well. The cells were incubated for an additional 2 h and the purple formazan formed by live cells was solubilized in dimethyl sulfoxide (DMSO) and the absorbance was colorimetrically measured at 570 nm. Cell viability was calculated considering the untreated group as 100% of viability.

### 2.10. Acute Lung Injury (ALI) Induction

Since a recent report did not find differences in the inflammatory response by comparing intratracheal and intranasal routes to induce ALI [18], we used intranasal administration of LPS (10 µg/40 µL) to induce ALI, as previous reported [19] with few modifications. For this, the mice were anesthetized (100 mg/Kg of ketamine; 10 mg/Kg of xylazine) and LPS diluted in sterile saline was administered. A group of mice was pretreated with EtOAc.f (30 mg/Kg, via oral) 24 and 1 h before ALI induction. After 24 h, the animals were euthanized and subsequent experimental procedures were performed.

### 2.11. Cytokine Levels and Cell Count in Bronchoalveolar Lavage Fluid (BALF)

To collect BALF, a cannula was inserted in the trachea and 0.6 mL of PBS containing EDTA (100 mM) was injected into the lung and subsequently aspirated. This procedure was repeated three times and, in the end, three aliquots were obtained. All aliquots were centrifuged (400× *g*, 10 min, 4 °C) and, the supernatant of the first aliquot was reserved to cytokine dosages through ELISA assay kit as aforementioned. The cell pellet was suspended in PBS/EDTA in a single aliquot, and the number of cells was determined by using optical microscopy and a Neubauer chamber.

### 2.12. Myeloperoxidase (MPO) Activity

Infiltration of activated phagocytes into lung tissue was estimated by assaying MPO activity following the previous description with slight modifications [20]. Lung tissues were homogenized in ice-cold solution (100 mM NaCl, 20 mM NaPO_4_, 15 mM Na-EDTA, pH 4.7) and then centrifuged (15,000× *g*, 15 min, 4 °C). Supernatants were removed and the pellet resuspended in potassium phosphate buffer (50 mM, pH 5.4) with 0.5% of H-TAB (hexadecyltrimethylammonium bromide) following new centrifugation under the same conditions described above. MPO activity was measured by incubating 10 µL of supernatant with TMB (1.6 mM) and H_2_O_2_ (0.5 mM) at 37 °C for 5 min. Changes in the optical density (O.D.) were monitored at 630 nm.

### 2.13. Tissue Processing and Protein Dosage

Lung tissue was homogenized in phosphate buffer (20 mM, pH 7.4) and then centrifuged at 1000× *g* for 10 min at 4 °C. Aliquots from the supernatant were used for each assay evaluating oxidative stress. Total protein content in the supernatant of tissue homogenate was determined through Bradford assay [21].

### 2.14. Determination of Superoxide Dismutase (SOD) and Catalase (CAT) Activities, GSH, and Protein Carbonylation Content

SOD activity was proportional to the ability of supernatant aliquots to inhibit pyrogallol (24 mM) autoxidation, which was monitored following the increase in absorbance at 420 nm [22]. Enzyme activity was calculated as inhibition percentage. CAT activity was evaluated by monitoring the reduction in absorbance at 240 nm resulting from hydrogen peroxide (0.2%) decomposition [23]. To measure GSH, content proteins present in the supernatant of tissue homogenates were precipitated with metaphosphoric acid (0.2 M). Non-protein fraction was used in the reaction with ortho-phthalaldehyde (1 mg/mL) and changes in fluorescence (350 nm excitation; 420 nm emission) were determined. GSH content was estimated by comparing to a standard curve [24]. Protein carbonyl content was determined by monitoring the formation of dinitrophenyl (DNP) resulting from the reaction of protein carbonyl and 2,4-dinitrophenylhydrazine (DNPH) as previously reported [25].

### 2.15. Histopathological Analysis

The lung tissue was collected, and following 72 h of fixation, it was processed (histotechnical equipment Lupetec, PT 05). After paraffin inclusion in paraplast (Leica paraplast), serial sections of 4 µm thickness from lung tissue were dewaxed at 60 °C and then stained with hematoxylin and eosin (H&E). The histopathological examination was performed using a light microscope (Leica, DMC 2900 coupled to a computer using the LAS software, Leica microsystem CMS GmbH). The specimens were evaluated through blind histological examination and classified according to a scoring system: lung morphology was classified as *preserved*, *partially preserved*, or *not preserved*; and inflammation was defined as *absent*, *discreet*, or *accentuated*.

### 2.16. In Vivo Toxicity Analysis of EtOAc.f

Mice received oral administration EtOAc.f (30 mg/Kg) 24 and 1 h before blood collection. The control group was treated with vehicle. Then, mice were anesthetized (100 mg/Kg of ketamine; 10 mg/Kg of xylazine) and a heparinized capillary was inserted into medial medial canthus of eye. The blood was gently transferred to a microtube after the volume required had been collected. The hematological analysis was performed in Auto Hematology Analyse BC-2800 (Mindray Biomedical Electronics, Nansham, China). Leukocyte differential analysis was assessed by a trained professional who examined a blood smear using optical microscopy. Next, the blood was centrifuged 3000 rpm for 7 min (L-1202 Spin^®^, Loccus) and plasma was collected. Aspartate aminotransferase (AST), alanine aminotransferase (ALT), creatine, and urea levels were determined by using commercial kit (Bioclin^®^, Quibasa-Química Básica, Belo Horizonte, Brazil) following the manufacturer’s instruction.

### 2.17. Statistical Analysis

The statistical analyses and graphics were done using GraphPad Prism 6.0 software. All analyses were performed in triplicate and the data were expressed as mean ± standard deviation. The significance of difference was calculated using one-way and two-way ANOVA, and Dunnett’s post-test for multiple comparisons. Values of *p* < 0.05 were considered significant.

## 3. Results

### 3.1. Inhibitory Effects of EtOAc.f and BuOH.f from A. muricata Leaves on Neutrophil-Derived ROS

To evaluate whether crude extract and fractions from *A. muricata* leaves could modulate ROS generation by phagocytic cells, we first screened these materials in an opsonized zymosan (Zy-Ops)-induced neutrophil ROS production assay. Neutrophils were pretreated for 30 min with different concentrations of EtOH.cr and each fraction. We found that EtOAc.f and BuOH.f had a greater effect on 3 µg/mL and 10 µg/mL (Table 1), and unlike other fractions, they demonstrated a better pharmacological profile in the concentration-response curve (Figure 1A–D).

Since EtOAc.f and BuOH.f presented more promising results, we evaluated if they had cytotoxic effects. Cell viability assay showed no cytotoxic effects of fractions at the highest concentration (10 µg/mL), (Figure 1E,F).

### 3.2. EtOAc and BuOH Fractions Reduce BMDMs ROS Generation

Based on our initial results, we decided to investigate the effects of EtOAc.f and BuOH.f as well on BMDMs. For this, the intermediate concentration of 3 µg/mL of each fraction was chosen for all subsequent experiments. In a similar experiment as described in Section 3.1, BMDMs pretreated with both EtOAc.f and BuOH.f produced lower ROS levels than untreated cells, following Zy-Ops stimulus (Figure 2A–D). Cytotoxic effects of the fractions on BMDMs were not observed in the cell viability assay (Figure 2E).

### 3.3. EtOAc and BuOH Fractions Modulate Inflammatory Functions of BMDMs

Zymosan particles are engulfed during phagocytosis, and this process triggers ROS generation by BMDMs [26]. To evaluate whether EtOAc.f and BuOH.f could modulate phagocytosis, we quantified BMDM engulfing Zy-FITC through flow cytometry assay. We found that EtOAc.f and BuOH.f pretreatment did not affect BMDMs-phagocytic capacity (Figure 3A,B). In addition, we used PMA (100 ng/mL) as a phagocytosis-independent stimulus to induce ROS generation by BMDMs. Our results showed that pretreatment of BMDMs with EtOAc.f and BuOH.f reduced ROS levels after PMA stimulus (Figure 3C,D).

When macrophages are exposed to an inflammatory stimulus, they produce different cytokines that can affect the function of different cell types. To evaluate whether fractions from *A. muricata* leaves could modulate cytokine secretion by BMDMs, the cells were pretreated with EtOAc.f and BuOH.f and then stimulated with LPS (1 µg/mL). It was found that BMDMs produced lower levels of IL-6 (Figure 3E) when submitted to pretreatment with EtOAc.f and BuOH.f. No effect on TNF-α secretion was observed (Figure 3F). The flow cytometry analysis of surface co-stimulatory molecules following LPS stimulus revealed that EtOAc.f and BuOH.f reduced expression of CD40 molecules by F4/80^+^ BMDMs (Figure 3G,H). No change in CD80 expression was observed (data not shown).

### 3.4. EtOAc Fraction Reduces MPO Activity and Recovers Oxidative Damage in the Lung during Experimental LPS-Induced ALI

The profile of chemical compounds present in BuOH.f and EtOAc.f has been recently reported by our group [15]. BuOH.f presents chlorogenic acid; procyanidin B2 and C1; (epi)catechin; quercetin (diglucoside, glucosyl-pentoside and xyloside-rutinoside); and rutin. In addition to all compounds found in BuOH.f, with the exception of quercetin-xyloside-rutinoside, EtOAc.f presents caffeic acid; quercetin and quercetin-rhamnoside; and kaempferol-rhamnoside. Thus, we decided to use only EtOAc.f to evaluate in vivo anti-inflammatory and antioxidant activities.

For the in vivo studies, we carried out a murine model of ALI induced by intranasal administration of LPS. C57BL/6 mice were pretreated with oral administration of EtOAc.f (30 mg/Kg) and challenged with LPS (10 µg/40 µL). Inflammatory and oxidative stress parameters were determined in the lung 24 h after LPS administration. The analysis of the number of cells in BALF revealed that LPS induces an intense leukocyte infiltration in the lung and EtOAc.f pretreatment has no effect on this parameter (Figure 4A). On the other hand, MPO activity was significantly lower in mice receiving EtOAc.f when compared to vehicle-treated mice (Figure 4B). Levels of TNF-α and IL-6 in BALF were also assessed and our results showed that EtOAc.f does not affect the cytokine secretion induced by LPS administration (Figure 4C,D).

Lung inflammation has been linked to dysfunction in the control of oxidative balance. In this regard, we also assessed oxidative stress markers in lung tissue. Our results showed that intranasal administration of LPS in mice was associated with an increase in protein carbonyl levels, which was significantly prevented by pretreatment with EtOAc.f (Figure 4E). To investigate changes in the enzymatic antioxidant system, we determined SOD and CAT activities. We found that the LPS challenge significantly reduced SOD and CAT activity and EtOAc.f pretreatment was unable to restore their activities (Figure 4F,G). Additionally, we determined GSH content and found that EtOAc.f has a protective effect on LPS-induced reduction of GSH levels (Figure 4H).

The histopathological analysis of lung from the vehicle group revealed a partially preserved simple epithelium and an increase in connective tissue with a consequent reduction in alveolar space; a hyaline material deposit associated with pulmonary edema and an accentuated inflammatory infiltrate were also observed (Figure 5C,D and Table 2).

On the other hand, in the EtOAc.f-treated group the connective tissue and pulmonary alveoulus were preserved, and only a discreet inflammation was identified (Figure 5E,F and Table 2). The saline group did not present any change in lung tissue (Figure 5A,B and Table 2).

Importantly, the oral treatment with EtOAc.f did not induce significant change in the plasma levels of hepatic and renal markers of lesion and no change in the hematologic profile of red blood cells, platelets, and myeloid compartment was observed by comparing to the control group (Appendix A).

### 3.5. EtOAc.f from A. muricata Leaves Did Not Produce Toxic Effects In Vivo

We evaluated if the oral treatment with EtOAc.f could promote toxicity in vivo. Importantly, we found that the oral treatment with EtOAc.f did not induce significant changes in the plasma levels of AST and ALT evaluated as a hepatic lesion marker and creatinine as a renal marker of lesion (Figure 6).

Moreover, no change in the hematologic profile of red blood cells, platelets, and myeloid compartment (Table 3) was observed by comparing to the control group.

## 4. Discussion

*Annona muricata* Linn. has been used in traditional medicine to treat different inflammatory disorders including respiratory conditions. However, only a few reports have addressed this topic. Our study provides new evidence that polyphenols-rich EtOAc.f and BuOH.f reduce the generation of ROS by neutrophils and macrophages and cause anti-inflammatory activity, reducing the secretion of IL-6 and expression of co-stimulatory molecule CD40. In LPS-induced ALI, EtOAc.f oral treatment reduced MPO activity and oxidative damage in the lung, confirming the in vitro anti-inflammatory and antioxidant activities of fractions of *A. muricata* leaves.

ROS are unstable molecules generated as by-products of oxygen metabolism and, at physiological levels, they have been implicated as regulators of cell signaling [27,28]. However, high levels of ROS induce damage to cellular components contributing to human pathologies, including ALI/ARDS [29,30,31]. We found that ethyl acetate (EtOAc.f) and n-butanol (BuOH.f) fractions from *A. muricata* leaves were more efficient in reducing ROS generation by neutrophils after Zy-Ops stimulus. Furthermore, similar results were observed in BMDMs. Thus, unlike many studies that focus on ROS scavenging activity assays such as ORAC, DPPH, and ferric reducing/antioxidant power (FRAP) [7,8,32] to demonstrate antioxidant actions of *A. muricata* leaf extract, our study presents its unprecedented antioxidant effects on neutrophils and macrophages.

Several studies have reported the in vitro cytotoxic activity of extracts of A. muricata leaves and different results have been described. In this regard, the final cytotoxic activity depends on the extract features and cell line used in experimental protocols [33]. In the present study, EtOAc.f and BuOH.f did not demonstrate cytotoxic action, suggesting that their antioxidant effects cannot be attributed to a reduction in cell viability.

Following recognition during phagocytosis, particles of Zy-Ops are internalized into the phagosome, inducing assembly and activation of nicotinamide adenine dinucleotide phosphate (NADPH) oxidase, a membrane-bound multimeric enzyme that promotes ROS generation [34]. The NADPH oxidase assembly is a crucial mechanism of ROS generation involved in antimicrobial activity of phagocytic cells [35] and, this enzyme has been a target in the development of therapies for ALI [36]. Regarding phagocytosis, both neutrophils and macrophages use similar mechanisms to internalize and promote ROS generation after the Zy-Ops stimulus. We found that Zy-FITC internalization was not modified by EtOAc.f and BuOH.f, indicating that their inhibitory effect on ROS production cannot be attributed to a reduction in phagocytosis of zymosan. However, EtOAc.f and BuOH.f attenuate PMA-induced ROS generation, suggesting an action on intracellular transduction pathways that leads to NADPH oxidase activation independently of phagosome formation. Indeed, it has been established that PMA, a nonphysiological soluble stimulus, promotes NADPH oxidase assembly by inducing protein kinase C (PKC) activation in a pathway of intracellular signal transduction that is downstream to phagosome formation [37].

Different signals contribute to macrophage activation resulting in the expression of surface molecules and secretion of pro-inflammatory cytokines. We noted that EtOAc.f and BuOH.f reduced CD40 expression and IL-6 secretion in BMDMs stimulated with LPS. CD40 is a member of the TNF receptor (TNFR) family, whose expression in macrophages has been associated with pathologic conditions [38], and IL-6 has been described as a crucial pleiotropic pro-inflammatory cytokine that regulates inflammation during cancer, autoimmunity, and infectious diseases [39]. Corroborating studies have demonstrated that quercetin, a flavonoid that was found in EtOAc.f and BuOH.f, reduced expression of CD40 in alveolar macrophages following stimulus with imiquimod, a ligand for the Toll-like receptor-7 [40]. Moreover, the phenolic compound cholorogenic acid, also found in both EtOAc.f and BuOH.f, suppresses IL-6 secretion in RAW 264.7 macrophages [41]. In this scenario, the regulatory activity of EtOAc.f and BuOH.f on macrophages may be beneficial in the management of inflammatory responses.

Recently, through mass spectrometry analysis, our group showed that BuOH.f from *A. muricata* leaves presents the same phytoconstituents found in EtOAc.f [15] (Appendix A). In fact, EtOAc.f and BuOH.f have a high level of phenolic compounds and some of these compounds were identified in both EtOAc.f and BuOH.f due to the similarity of polarity between the solvents. However, quercetin-pentoside and quercetin-rhamnoside were identified only in EtOAc.f. Quercetin-xyloside-rutinoside was identified only in BuOH.f. Since ethyl acetate was used for extraction prior to n-butanol, EtOAc.f concentrated the phenolic compounds that were also extracted by the similar polarity n-butanol solvent. It is worth mentioning that the content of total polyphenols was higher in EtOAc.f [15]. This corroborates other studies done by our research group that showed similar phytochemical makeup between the ethyl acetate and n-butanol fractions [42].

Since our in vitro studies showed a similar profile of action between EtOAc.f and BuOh.f, we determined only the relevant effects of EtOAc.f in the animal model of LPS-induced ALI that replicate pathologic features of human lung disease such as endothelial injury, leukocyte infiltration, and edema [43]. We found that EtOAc.f pretreatment reduced MPO activity in the lungs, and these results corroborate with a recent study reporting that topical application of aqueous extract of *A. muricata* leaves reduced MPO activity in the skin during cutaneous inflammation [44]. Since MPO is expressed primarily in azurophilic granules of neutrophils [45], a reduction in the activity of this enzyme could suggest a specific action on neutrophil infiltration. Interestingly, we did not observe a reduction in the total number of leukocytes in BALF. These conflicting results may be attributable to the fact that, in addition to neutrophils, BALF includes other cell types such as macrophages and lymphocytes [46], and the global count of cells in BALF could hide a specific reduction in the neutrophil population.

Studies have suggested a crucial role of macrophages in ALI [47,48] and therapies directed against macrophage-mediated inflammation opens new opportunities to treat lung inflammation. The activation of macrophages leads to the secretion of pro-inflammatory mediators such as cytokines and nitric oxide (NO). IL-1β, TNF-α, and, IL-6 are secreted in response to LPS and they exacerbate the inflammatory response and induce neutrophil recruitment [48]. High levels of IL-6 and TNF-α have been associated with worse prognosis and higher mortality rate in patients with lung disease [49,50]. In our experimental conditions, EtOAc.f pretreatment did not change the levels of IL-6 and TNF-α in BALF. Although these results could indicate the total absence of effect of EtOAc.f on levels of IL-6 and TNF-α in vivo, they also suggest that other cells may contribute to the secretion of cytokines in the lungs following an LPS challenge. Indeed, one study revealed the constitutive expression of the IL-6 gene in lung epithelial cells from naïve mice, and the exposition of these cells to fungal-allergen promoted IL-6 secretion [51]. Moreover, lung epithelial cells have been also described as a source of TNF-α [52]. We also determined the levels of NO in BMDMs and lung tissue, where we found that EtOAc pretreatment did not affect NO levels after an LPS challenge, both in vitro and in vivo (data not shown). These latter results suggest that the EtOAc.f acts by mechanisms independent of NO production.

The interplay between oxidative stress, inflammation, and lung diseases has been reported [53]. Patients who survived ARDS present lower levels of oxidized proteins in BALF when compared to non-survivors, suggesting that oxidative damage may indicate a worse prognosis [54]. The oxidative stress with protein oxidation and inflammatory response is also evident in LPS-induced ALI [55]. Only a few studies have associated the in vitro to in vivo antioxidant activities of *A. muricata*, and reports showing its effects on experimental lung inflammation are currently lacking. Here, we demonstrated that oral administration of 30 mg/Kg of EtOAc.f reduced oxidative damage in the lungs by decreasing protein carbonylation levels. This effect was associated with protection from LPS-induced reduction of GSH levels, suggesting an action on the glutathione system. This is very relevant since GSH supplementation presented protective effects on oxidative stress and mitochondrial function following experimental ALI [56]. Moreover, studies have confirmed that *A. muricata* is a source of compounds with protective activity on GSH content, since treatment with aqueous extract of leaves restores hepatic GSH levels of diabetic rats [57] and commercial dry extract protects from diabetes-induced reduction of testicular GSH levels [58]. Considering that EtOAc.f has remarkable antioxidant effects on neutrophils and BMDMs in vitro, it is plausible to propose that the in vivo action of EtOAc.f reducing oxidative damage in the lung could result from a decreased production of ROS by activated macrophages and neutrophils that reside or migrate to the lung. Indeed, studies have associated phagocyte-derived ROS with acute inflammatory response and tissue injury in the lung [59], and in this context, regulating ROS generation by phagocytes during lung inflammation may be a strategy to preserve the organ function.

It is important to emphasize that although the effects of EtOAc.f and BuOH.f might reflect the action of the set of compounds present in each fraction, corroborating reports have shown that isolated compounds found in these fractions demonstrated protective effects in lung inflammation. For instance, oral administration of quercetin reduced lung inflammation, depending on the cAMP-PKA/exchange protein activated by the cAMP-1 (Epac-1) pathway [60]. Moreover, inflammatory features such as tissue injury and cell infiltration in the lung induced by intraperitoneal administration of LPS were attenuated by nasogastric treatment with epicatechin [61]. Although more detailed studies are needed, the herein presented purpose of using fractions of *A. muricata* leaves can offer some advantages: the procedures for preparing EtOAc fractions are less complex when compared to the purification of isolated compounds; the possibility of oral administration, which is less expensive, more convenient, and acceptable for human patients; and oral use did not cause hepatic, renal and hematological toxicity, showing the safety of the treatment.

## 5. Conclusions

In conclusion, EtOAc.f and BuOH.f from *A. muricata* leaves contain bioactive compounds that modulate pro-inflammatory markers and ROS generation in neutrophils and macrophages. Moreover, the EtOAc.f activities found in immune cell-based studies were associated with antioxidant and anti-inflammatory effects in vivo in a model of LPS-induced lung inflammation. The current study also reveals novel immunomodulatory activities of bioactive compounds present in *A. muricata* leaves and introduces new possibilities for the development of therapies for lung inflammation and other inflammatory disorders.

## Figures and Tables

**Figure 1 pharmaceutics-14-01182-f001:**
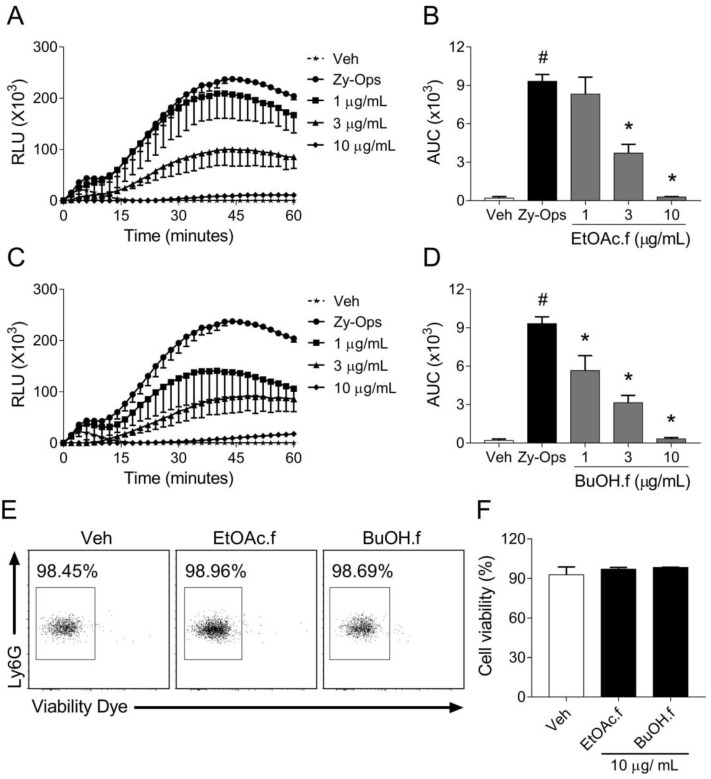
EtOAc.f and BuOH.f reduce ROS generation by neutrophils stimulated with opsonized zymosan. (**A**–**D**) Concentration-response curves of EtOAc.f (**A**,**B**) and BuOH.f (**C**,**D**) on the production of ROS by neutrophils in response to Zy-Ops (100 µg/mL); (**A**,**C**) are representative curves of ROS production over time; and (**B**,**D**) represent the area under the curve (AUC) as quantification of ROS. Each group was analyzed in triplicates. (**E**,**F**) Cell viability was determined through flow cytometry assay using Viability Dye; (**E**) representative dot blot resulting from cell acquisition in flow cytometer; (**F**) cell viability was calculated as the percentage of untreated cell group (Veh). Data represent mean ± S.E.M. of triplicates of one representative experiment. # and * indicate a statistically significant difference comparing to Veh group or Zy-Ops group, respectively, *p* < 0.05.

**Figure 2 pharmaceutics-14-01182-f002:**
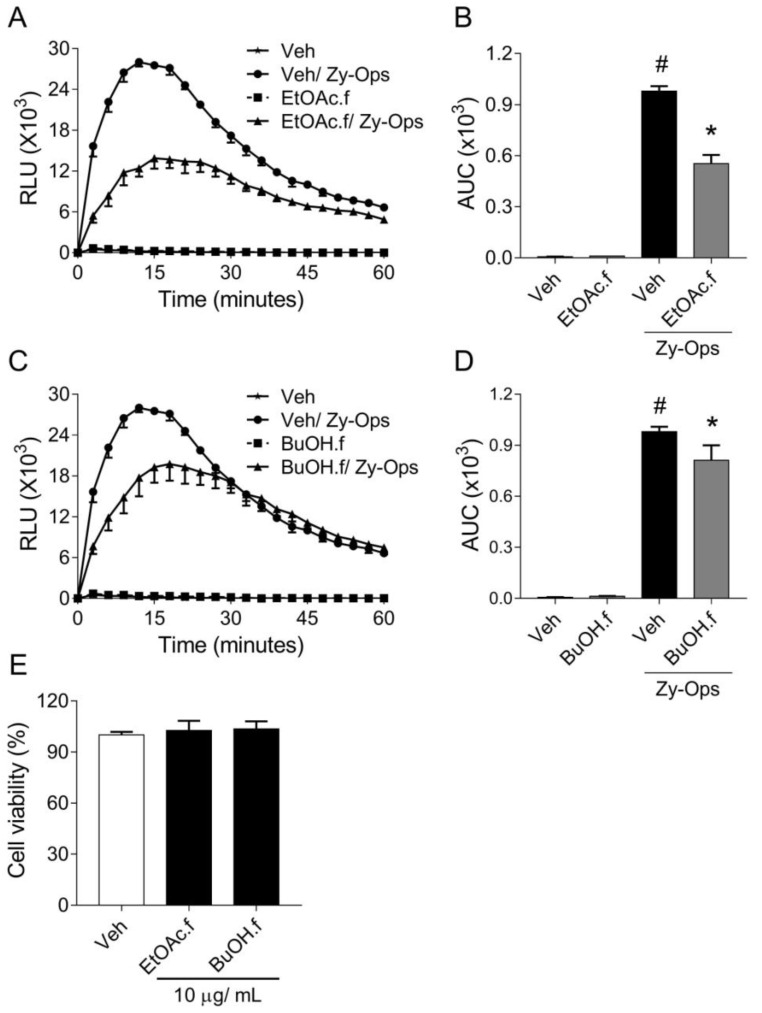
Pretreatment with EtOAc.f and BuOH.f reduces opsonized zymosan-induced production of ROS by BMDMs. (**A**–**D**) BMDMs were pretreated for 30 min with 10 µg/mL of EtOAc.f or BuOH.f following Zy-Ops stimulus (100 µg/mL); (**A**,**C**) are representative curves of ROS production over time; and (**B**,**D**) represent the area under the curve (AUC) as quantification of ROS. Each group was analyzed in triplicates. (**E**) Cell viability determined through MTT assay calculated as the percentage of untreated cell group (Veh). Data represent mean ± S.E.M. of triplicates of one representative experiment. # and * indicate statistically significant difference comparing to Veh group or Zy-Ops group, respectively, *p* < 0.05.

**Figure 3 pharmaceutics-14-01182-f003:**
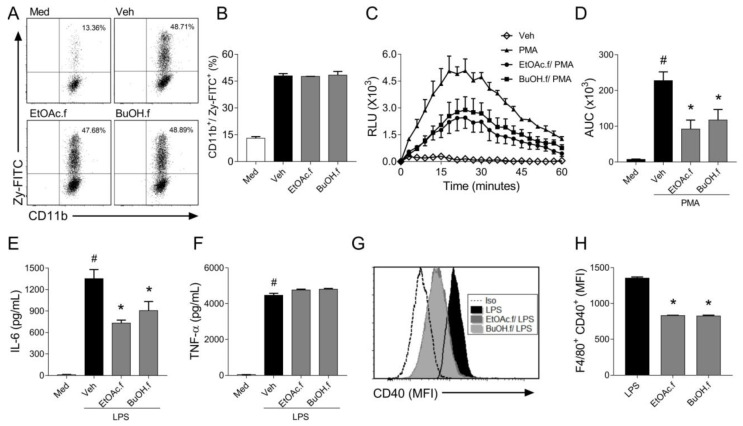
EtOAc.f and BuOH.f modulate inflammatory responses of BMDMs. BMDMs were pretreated (30 min) with 10 µg/mL of EtOAc.f or BuOH.f and inflammatory responses were evaluated. (**A**,**B**) BMDMs were incubated with FITC-labeled zymosan (100 µg/mL) and stained with CD11b. Phagocytic activity was assessed using a flow cytometer analyzing frequencies of CD11b^+^/Zy-FITC^+^ cells. (**A**) Representative dot plot from cell acquisition in a flow cytometer; (**B**) quantification of percentages of cells CD11b^+^/Zy-FITC^+^. (**C**,**D**) BMDMs were stimulated with PMA (100 ng/mL) and ROS production was monitored. (**C**) Curves of ROS production over time and (**D**) quantification of the area under the curve (AUC). (**E**) IL-6 and (**F**) TNF-α levels were determined in supernatant of culture of BMDMs stimulated with LPS (1 µg/mL) for 24 h (**G**,**H**). Analysis of CD40 expression on the surface of BMDMs stimulated with LPS. (**G**) Representative flow cytometry-based analysis showing Median Fluorescence Intensity (MFI) and (**H**) quantification of MFI as the expression of CD40 on F4/80^+^ macrophages. Data represent mean ± S.E.M. of triplicates of one representative experiment. # and * indicate a statistically significant difference comparing to Veh group or Zym group, respectively, *p* < 0.05.

**Figure 4 pharmaceutics-14-01182-f004:**
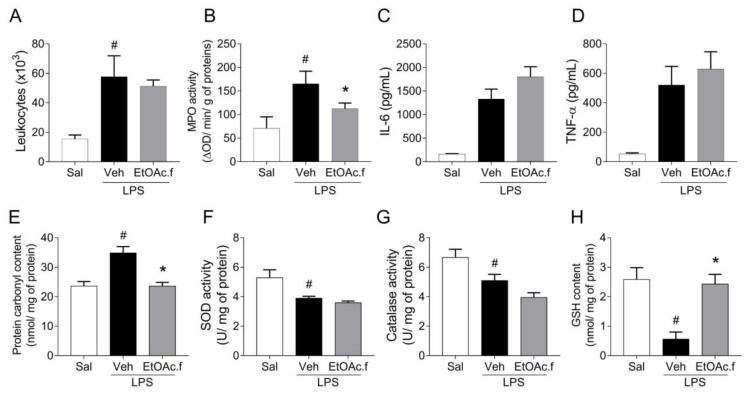
EtOAc.f presents anti-inflammatory and antioxidant activities in LPS-induced acute lung injury. Mice were pretreated with EtOAc.f (30 µg/mL, via oral) 24 h and 1 h before intranasal LPS (10 µg/40 µL) challenge and then, at 24 h after LPS administration, inflammatory (**A**–**D**) and oxidative (**E**–**H**) markers were assessed. (**A**) The total number of leukocytes present in BALF. (**B**) MPO activity evaluated in lung tissue. (**C**) Levels of IL-6 and (**D**) TNF-α determined in BALF through ELISA assay. (**E**) Protein carbonyl, (**F**) superoxide dismutase, and (**G**) catalase activity. (**H**) glutathione content was assayed in lung tissue homogenates. Data represent mean ± S.E.M. of one representative experiment (*n* = 5 mice/group). # and * indicate a statistically significant difference comparing to saline (Sal) group or vehicle (Veh) group, respectively; *p* < 0.05.

**Figure 5 pharmaceutics-14-01182-f005:**
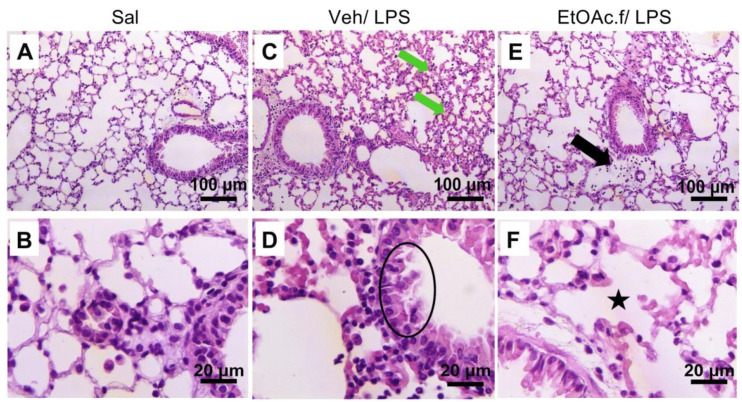
H&E staining of lung tissue sections. Histopathological features of the lung from saline (Sal), vehicle (Veh/LPS), and ethyl acetate fraction (EtOAc.f/LPS) groups. The mice were pretreated with vehicle or EtOAc.f 24 h and 1 h before LPS (10 µg/40 µL) challenge. Lung tissue was harvested 24 h after LPS and then processed for H&E staining. (**A**,**B**) No changes were observed in the Sal group. (**C**) In the Veh/LPS group, hyaline material deposit and (**D**) partially preserved simple epithelium are represented by green arrows and black circle, respectively. (**E**) Discreet inflammatory infiltrate (black arrow) and (**F**) intra alveolar space restored (black star) was evidenced in EtOAc.f/LPS group.

**Figure 6 pharmaceutics-14-01182-f006:**
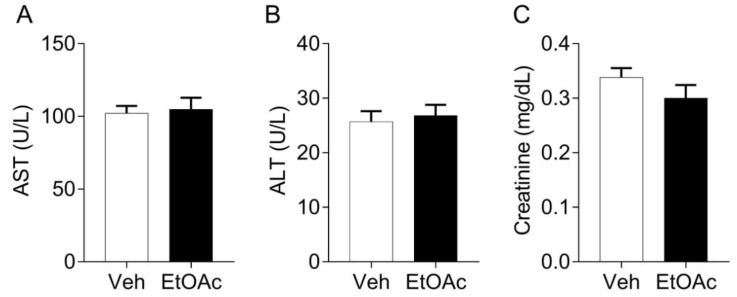
EtOAc.f does not change the plasma levels of biomarkers of toxicity. Mice were pre-treated with EtOAc.f (30 mg/kg, via oral) 24 and 1 h before whole blood collection. (**A**) AST, (**B**) ALT, and (**C**) creatinine plasma levels following EtOAc.f oral pre-treatment. Data represent mean ± S.E.M. (*n* = 5 mice/group).

**Table 1 pharmaceutics-14-01182-t001:** Screening of ethanol crude extract and fractions from *A. muricata* L. leaves on ROS production by neutrophils stimulated with opsonized zymosan. Neutrophils were purified from the bone marrow of C57/BL6 mice through Percoll density gradient. EtOH.cr extract and fractions from *A. muricata* L. leaves were prepared following the liquid-liquid partition method. In a concentration-response curve, 2 × 10^5^ neutrophils were pretreated for 30 min with different concentrations of each fraction, and then the cells were stimulated with Zy-Ops. ROS production was monitored through chemiluminescence emission of luminol for 60 min. The percentage of ROS production was calculated considering Zy-Ops stimulated cells without any pretreatment as 100%. Data represent mean ± S.E.M.

Fraction	ROS Generation (Percentage of Control) ± S.E.M.		
	Concentration		
	1 µg/mL	3 µg/mL	10 µg/mL
Ethanol (EtOHcr)	94.09 ± 6.92	92.20 ± 5.07	50.07 ± 2.17
Hexane (Hex.f)	140.93 ± 7.43	60.77 ± 10.54	57.45 ± 10.15
Dichloromethane (DCM.f)	59.66 ± 15.87	96.84 ± 4.24	35.33 ± 6.83
Ethyl Acetate (EtOAc.f)	89.41 ± 14.10	39.91 ± 7.23	3.11 ± 0.32
*n*-Butanol (BuOH.f)	60.67 ± 12.61	33.71 ± 6.21	3.44 ± 1.16
Water (Wa.f)	76.62 ± 8.64	50.77 ± 11.07	82.74 ± 19.83

**Table 2 pharmaceutics-14-01182-t002:** EtOAc.f protects against LPS-induced histopathological changes in the lung. Mice were pretreated with oral administration of vehicle (Veh) or EtOAc.f 24 h and 1 h before intranasal challenge with LPS (10 µg/40 µL). Following 24 h after LPS instillation, the qualitative morphological analysis of each lung structure was classified as *preserved*, *partially preserved*, or *not preserved*; inflammation was defined as *absent*, *discreet*, or *accentuated*.

Histopathology	Sal	Veh/LPS	EtOAc.f/LPS
pseudostratified epithelium	preserved	preserved	preserved
simple epithelium	preserved	partially preserved	partially preserved
connective tissue	preserved	partially preserved	preserved
pulmonary alveolus	preserved	partially preserved	preserved
inflammation	absent	accentuated	discreet

**Table 3 pharmaceutics-14-01182-t003:** Oral treatment with EtOAc.f did not induce changes in hematological profile of cells. Mice were treated with EtOAc.f (30 mg/Kg) or Vehicle (Veh) and comparative hematological analysis between groups was performed. WBC (White Blood Cells); RBC (Red Blood Cells); HCT (Hematocrit); Hb (Hemoglobin); MCH (Mean Corpuscular Hemoglobin); PL (Platelets); ND (not detected). Data represent mean ± S.E.M. (*n* = 5 mice/group).

	Veh	EtOAc
WBC (×10^3^/μL)	13.92 ± 2.78	13.24 ± 2.11
Neutrophil (%)	14.6 ± 0.97	13.00 ± 0.83
Eosinophils (%)	4.00 ± 0.83	3.60 ± 0.67
Basophils (%)	ND	ND
Lymphocytes (%)	77.6 ± 1.69	81.00 ± 1.14
Monocytes (%)	2.00 ± 0.51	2.40 ± 0.81
RBC (×10^3^/μL)	8.94 ± 0.12	8.73 ± 0.14
HCT (%)	40.64 ± 0.35	39.98 ± 0.41
Hb (g/dL)	14.02 ± 0.14	13.62 ± 0.17
MCH (pg)	15.72 ± 0.08	15.56 ± 0.11
PL (×10^3^/μL)	661.00 ± 160.50	775.8 ± 195.90

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
