# Peer review of "Polyphenols-Rich Fraction from Annona muricata Linn. Leaves Attenuates Oxidative and Inflammatory Responses in Neutrophils, Macrophages, and Experimental Lung Injury"

_pharmaceutics, 2022, doi:10.3390/pharmaceutics14061182_

Round 1

Reviewer 1 Report

The experimental data and manuscript preparation are good. However, the English of the manuscript is bad. Please ask an local English speaker to help you improve it.

Reviewer 2 Report

Page 3.

Line 96. Authors obtained the different fractions by using solvent-solvent partition method. Please describe the volume ratio of solvent in partition.

Line 107. The Retention time of compound 3, 4, and 5 in HPLC chromatogram (Fig S1) and supplementary Table 1 are not consistent. Please check and revise.

Page 6. Table 1.

There are 6 fractions obtained from A. muricate L. leaves. Among these fractions, the EtOAC.f and BuOH.f were shown more promising results.

  • I would suggest authors present the HPLC-MSMS data of the 2 fractions, not only the EtOAC.f fraction.
  • Please compare the EtOAc.f and BuOH.f chromatographic profile.
  • Please explain why the different solvent polarity of EtOAc and BuOH can obtain the similar (almost the same) extract constituents as authors described in Page 12. Line 426.

Page 11 Figure 5.

  • Authors use capital letter in figure legend but use lower case in figure.
  • The “red” arrow is not clear in purple background figure.

Reviewer 3 Report

The manuscript submitted by Andre et al et al describes the phenomenon, Polyphenols rich fraction from Annona muricata Linn. leaves attenuate s oxidative and inflammatory responses on neutro-phils and macrophages and experimental lung injury. The manuscript is interesting and eventually contains relevant information for readers, However, there are many points to be addressed by the authors.

The Authors demonstrated that this plant extract can prevent activation of neutrophils and macrophage to secrete cytokines. Is this phenomenon limited to these cell types or does it extend to other cells such as stromal and endothelial compartments? Also, after LPS stimulation, the expression of IL-6, TNFa, and CD40 was reduced; what is the relationship between these factors? Although there is crosstalk between all of these factors, each factor has its own receptors and pathway, so the authors should explain how this works mechanistically.

Authors can conduct a rescue experiment by administering recombinant IL-6, TNFa, or restoring CD40 by overexpression and administering the extract to see what happens.

How about combination experiments extract and neutralizing antibodies against these factors (IL-6, TNFa and CD40). How about testing this extract in a pure system, such as a cell line like RAW or THP-1, and seeing if the same thing happens?

In vivo

What happened to the mice's blood counts (RBCs, platelets, and myeloid compartments) and bone marrow count, and if possible, show hematoxylin and eosin staining for BM?

Is this extract toxic in vivo and in vitro at high doses? What happened to the mCSF ligand and receptors? Because the author mentioned that he stimulated macrophages with a condition medium containing this ligand, I'm concerned about the expression of MCSF and its receptors on macrophages after extract treatment. Does the extract inhibit MCSF in vivo? Does the extract work the same in other inflammation models as it does in the ALI model?

Why did the authors not quantify the chemical composition of BuOH.f and EtOAc.f, which had a greater effect at 3 mg/mL and 10 mg/mL, respectively? should explain why the BuOH fraction has the greatest effect despite having lower flavonoids and phenols contents.

According to the authors' interpretation of the EtOAc extract HPLC graph, Catechin was eluted before Quercetin-diglucoside, Quercetin-glucosyl-pentoside, and Quercetin-glucoside, despite the fact that all of these glucosides are more polar than Catechin. Need justifications.
